# Peer review of "Pedigree-Based Genetic Diversity in the South African Boerboel Dog Breed"

_animals, 2024, doi:10.3390/ani14060975_

Round 1

Reviewer 1 Report

Comments and Suggestions for Authors

Dear authors, 

Thank you for submitting interesting paper. I think you did a good job in analyzing genetic diversity in BBD. However, I have several suggestion that might improve the paper. The most important is related to additional measures of genetic diversity measures: the coefficient of inbreeding is a good measure, but inbreeding rate and effective population size might reveal more about the loss of genetic diversity. 

The next suggestion is related to the practical implications of the paper. Please improve discussion and conclusion by adding lines explaining how will obtained results influence breeding strategies in practical sense. 

Finally, I suggest that authors put the results of the pedigree analysis in the context of genomic era and future studies. 

Author Response

Dear Reviewer,

The authors would like to thank you for your invaluable inputs aimed at enhancing clarity and readability of our manuscript.

We are greatly appreciative of your support

We trust that the authors have addressed your valuable edits to improve our manuscript.

Herewith below please find the detailed responses to your specific comments:

Comment

Thank you for submitting interesting paper. I think you did a good job in analyzing genetic diversity in BBD. However, I have several suggestion that might improve the paper. The most important is related to additional measures of genetic diversity measures: the coefficient of inbreeding is a good measure, but inbreeding rate and effective population size might reveal more about the loss of genetic diversity.

Answer

Results

Line 208-239: Inbreeding rate and effective population size

We have included Figure 2 of Average inbreeding coefficient by year of birth in the Boerboel dog breed in the manuscript text to present the average inbreeding over the pedigree period.

We showed that Inbreeding increased steadily from 0.047 to 0.091 over the period 1983 and 2019, respectively. The rate of increase per year of 0.20% falls at the borderline to ensure sufficient genetic variability [29] (Weigel, 2001). The observed rate of inbreeding was lower than reported for the South African Boxer breed (0.42%) over the period 1960 and 1980 [30] (Mostert et al., 2015). The rates of increase in inbreeding in both South African dog breeds were however, lower than the critical rate of increase of 0.5% [29,31] (FAO, 1998; Weigel, 2001).

The effective population size, which measures the amount of genetic variability within a population, calculated from the individual inbreeding rate at 83.1, was within the recommended size [32,33] (Meuwissen, 1997; Sorensen et al., 2005).

Line 284-292: Discussion

The observed rate of inbreeding per year of 0.20% was within the acceptable range of 0.5% per year or 0.5 to 1% by Food and Agriculture Organization [31] (FAO, 1998). Similarly, the effective population size observed in the BBD (83.1) was within the recommended range of 50-100 animals, allowing for a 1% increase in inbreeding rate per generation. Meanwhile, Meuwissen and Woolliams (1994) [49] suggested Ne of 31-250 for maintaining population fitness. Even though the Ne in the BBD is clear of critical or danger rates, it is recommended that management promotes improving of genetic diversity through ensuring an equal ratio of males and females in mating programs [50] (Willi et al., 2022) and adding more genetically distant individuals.

Comment

The next suggestion is related to the practical implications of the paper. Please improve discussion and conclusion by adding lines explaining how will obtained results influence breeding strategies

in practical sense.         

Answer

Line 362-372: Our research has established that genetic diversity in the BBD is low but not eroded and at risk of extinction. The BBD may possess unique survival genetic variants of the future. Thus, breeding programmes need to maintain this diversity for continued survival of the BBD and to assist in restoring the diversity of commercial breeds likely to lose diversity through intense selection. Diversity management strategies could identify a reasonable amount of animals of genetic value that need to be retained, and with sufficient females underlying the population growth rate [61] (Fernádez et al., 2011). Furthermore, management strategies need to ensure equal genetic contribution of parents to future generations to maximise the observed total genetic variation being explained by a handful of animals. Hierarchical designs and optimum contribution strategies have been explored as a way to balance parental contributions to the population [61] (Fernádez et al., 2011).

Comment

Finally, I suggest that authors put the results of the pedigree analysis in the context of genomic era and future studies.

Answer

Line 373-382: Pedigree analyses are useful resources for providing a better understanding of the populations’ history, allowing for the identification of historical events, such as the founder animals and genetic bottlenecks [72] (Valera et al., 2005). Although pedigree data are important for determining genetic diversity, it is limited by the start of recording, missing and incorrect recording [20] (Leroy, 2011). Availability of genomic data has facilitated landscape genomics and conservation studies to accurately estimate genetic diversity and inbreeding [8,21,22] (Mäki et al., 2010; Gajaweera et al., 2019, Radko & Podbielska, 2021). Genomic data enable accurate assessment of the population structure, based on actual versus expected genetic diversity [23] (Gonzalez-Cano et al., 2022). Genomic information on the other hand is still limited by the cost of marker genotyping.

Reviewer 2 Report

Comments and Suggestions for Authors

This paper reports the findings of analyses on inbreeding levels in the South African Boerboel dog breed. Those analyses are sound and the results will be useful to conserve the breed into the future. There are a few minor editorial type suggestions made in chronological order in the dot points below and these should be attended to before the paper is accepted for publication.

·             Line 37 – change the beginning of the sentence to read “The numbers of founders were …” or “The number of founders was …” (i.e. need to match either singular or plural use)

·             Line 76 – change “affect” to “affects”

·             Line 80 – insert “the” between “assessing” and “dog breed”

·             Lined 83-84 – rather than exclude them from mating, could they be joined to totally unrelated animals or is this recommendation more related to the number of founder animals?

·             Line 103 – delete “of” from this line

·             Line 107 – suggest change to read “parents appeared before offspring, there were no duplications …”

·             Line 140 – change “analysis” to “analyses”

·             Line 143 – delete “have”

·             Lines 146 – change “represent” to “represents”

·             Line 221 – change “was” to “were”

·             Lines 252-253 – suggest change to “… maintain the genetic diversity, while managing inbreeding levels to ensure they do not affect future selection …”

·             Line 260 -  insert “the” between “since” and “male-to-female”

·             Lines 262 and 268 – change “Pe-rez” to “Perez”

·             Line 264 – suggest change to read “… was used equally over generations.”

·             Line 274 – suggest change to read “… but these were lower …”

·             Line 293 – suggest change to read “Similar ratios were shared …”

·             Line 295 – suggest change “contribution” to “contributions”

·             Line 303 – suggest delete comma after “2011)” and replace with “and”

·             Line 306 – change “ration” to “ratio”

·             Line 314 – suggest change to read “… required to monitor …”

·             Line 315 – delete “monitor”

·             Line 329 – suggest change to read “… an effective way to increase the genetic diversity.”

Comments on the Quality of English Language

See minor editorial suggestions in previous section

Author Response

Dear Reviewer,

The authors would like to thank you for your invaluable inputs aimed at enhancing clarity and readability of our manuscript.

We are greatly appreciative of your support

We trust that the authors have addressed your valuable edits to improve our manuscript.

Herewith below please find the detailed responses to your specific comments:

Comment

Line 37 - change the beginning of the sentence to read 'The numbers of founders were ... " or "The number of founders was ... " (i.e. need to match either singular or plural use)

Answer

Line 37 - The number of founders () were 348 and 356, and the effective number of founders () 57.4 and 60.1 for males and females were observed, respectively

Comment

Line 76 - change "affect" to "affects"

Answer

Line 76 - Because pedigree dogs are kept in small, closed groups without external gene flow and very few dogs are used for breeding [8] (Mäki, 2010), inbreeding consequently affects their performance, reproductive qualities, and adaptability [9] (Martínez et al., 2012).

Comment

Line 80 - insert "the" between "assessing" and "dog breed"

Answer

Line 80 - Characterization of population structure within breeds is required for assessing the dog breed's future reproductive potential [11] (Mokhtari et al., 2015), and monitoring gene flow by providing information on the population's founding ancestors and their contributions to the existing population's variability [12] (Stachowicz et al., 2018).

Comment

Lined 83-84 - rather than exclude them from mating, could

they be joined to totally unrelated animals or is this recommendation more related to the number of founder animals?

Answer

Lined 83-84 This can prevent the passing down of harmful recessive genes, genetic diseases such as hip and elbow dysplasia (Jansson, 2014; Ólafsdóttir Kristjánsson, 2008), and phenotypic defects that can reduce fitness and adaptation in offspring (Vostrá-Vydrová et al., 2016).

Comment

Line 103 - delete "of' from this line

Answer

Line 103 - These included 87 808 animal records, constituting 43 682 males and 44 084 females, as well as 42 animals without sex category records.

Comment

Line 107 - suggest change to read "parents appeared before offspring, there were no duplications ... "

Answer

Line 107 - The original data were edited to ensure parents appeared before offspring, there were no duplications, and was also organized to remain with animal identification, sire, dam, birth date, sex, and breeder identification as required by the software used for analysis.

Comment

Line 140 - change "analysis" to "analyses"

Answer

Line 140 - In this study, the RP was defined as animals born between 2009 and 2019, which included 36 546 animals (18 196 males and 18 350 females), where analyses for males and females were performed separately.

Comment

Line 143 - delete "have"

Answer

Line 143 - The sub-program prob_orig.f of the method developed by Boichard et al. (1997) [20] was used to trace back the most important ancestors that contributed to the genetic diversity of the afore-mentioned animals in the RP.

Comment

Lines 146 - change "represent" to "represents"

Answer

Line 146 - The estimated cumulative genetic contribution represents the total genetic variation explained by the population's most important ancestors.

Comment

Line 221 - change "was" to "were"

Answer

Line 221 - Pedigree with similar EqG levels was used by Mäki (2010) [8] for Nova Scotia and a slightly deeper pedigree was used in the Czech Spotted Dog by Machová et al. (2020) [26].

Comment

Lines 252-253 - suggest change to "... maintain the genetic diversity, while managing inbreeding levels to ensure they do not affect future selection ... "

Answer

Lines 252-253 - Therefore, evaluating the genetic structure of the BBD population is important to establish and maintain the genetic diversity, while managing inbreeding levels to ensure that they do not affect future selection [49] (Makina et al., 2014).

Comment

Line 260 - insert "the" between "since" and "male-to­ female"

Answer

Line 260 - According to Fernandes et al. (2010) [28], males often contribute more genes to the population since the male-to-female breeding ratio is greater than one.

Comment

Lines 262 and 268 - change "Pe-rez" to "Perez"

Answer

Lines 262 - This implies that a male is used to mate with a large number of females, similar to the polygynous mating system described for wild boar (Sus scrofa) [50] (Pérez-González et al., 2014).

Line 268 - As a result, both sexes should transfer significant amounts of the genes they have in order to maintain diversity [50] (Pérez-González et al., 2014).

Comment

Line 264 - suggest change to read "... was used equally over generations."

Answer

Line 264 - The findings of this study, however, indicate that the number of breeding males and females was used equally over generations.

Comment

Line 274 - suggest change to read "... but these were lower ..."

Answer

Line 274 - Similar findings were reported for the Dogue de Bordeaux dogs [45] (Leroy et al., 2006) and Bracco Italiano dogs [42] (Cecchi et al., 2013), but these were lower than ancestors in the Deutsch Drahthaar dogs [55] (Michels & Distl, 2022).

Comment

Line 293 - suggest change to read "Similar ratios were shared ..."

Answer

Line 293 - Similar ratios were shared by Shariflou et al. (2011) [24] for the Australian cattle dog and Border collie, indicating a loss of genetic diversity due to the use of the same animals as parents or unequal contributions of ancestor genes in the population [11,20,63] (Boichard et al., 1997; Mokhtari et al., 2015; Sheikhlou & Abbasi, 2016).

Comment

Line 295 - suggest change "contribution" to "contributions"

Answer

Line 295 - Similar ratios were shared by Shariflou et al. (2011) [24] for the Australian cattle dog and Border collie, indicating a loss of genetic diversity due to the use of the same animals as parents or unequal contributions of ancestor genes in the population [11,20,63] (Boichard et al., 1997; Mokhtari et al., 2015; Sheikhlou & Abbasi, 2016).

Comment

Line 303 - suggest delete comma after "2011)" and replace with "and"

Answer

Line 303 - These findings are similar to those found for Hannoverian hounds [68] (Voges & Distl, 2009), Bichon frise [24] (Shariflou et al., 2011), Australian shepherd, Papillon, and Phalene [48] (Wijnrocx et al., 2016) dog breeds.

Comment

Line 306 - change "ration" to "ratio"

Answer

Line 306 - The ratio in the BBD of South Africa could also be explained by the population's closed herd book, strong selection for appearance, and favouritism of some few individuals [37] (Ács et al., 2019).

Comment

Line 314 - suggest change to read"... required to monitor

Answer

Line 314 - Conservation strategies are required to monitor genetic diversity and inbreeding e.g. new selection programmes to increase the gene pool by mating sires that have not been mated before [62] (Bozzi et al., 2006), or genetically distant dog breeds.

Comment

Line 315- delete "monitor"

Answer

Line 315 - Conservation strategies are required to monitor genetic diversity and inbreeding e.g. new selection programmes to increase the gene pool by mating sires that have not been mated before [62] (Bozzi et al., 2006), or genetically distant dog breeds.

Comment

Line 329 - suggest change to read "... an effective way to increase the genetic diversity."

Answer

Line 329 - The introduction of new genetic material from genetically distant dog breeds could be an effective way to increase the genetic diversity.

Reviewer 3 Report

Comments and Suggestions for Authors

This is a sound study and is well-explained. One suggestion is to include a few details about "kinship," which is likely more important than inbreeding coefficients as a way to assure diversity. A few editorial comments:

Line 68. “purebred” instead of “thoroughbred.” In most situations Thoroughbred is a specific horse breed.

Line 84. “kinship” may be more important than “inbreeding. If a dog has a high inbreeding coefficient but a kinship of 0 with some other animals, the coefficient can be taken back to 0 by using that mating. It would be useful, if possible, to include this in the analysis, but this is an optional recommendation.

303. “Bichon Frise” for “Bichon fries”

Author Response

Dear Reviewer,

The authors would like to thank you for your invaluable inputs aimed at enhancing clarity and readability of our manuscript.

We are greatly appreciative of your support

We trust that the authors have addressed your valuable edits to improve our manuscript.

Herewith below please find the detailed responses to your specific comments:

Comment

Line 68. "purebred" instead of "thoroughbred." In most situations Thoroughbred is a specific horse breed.

Answer

Line 68 - However, a conflict exists between the SABBS recognizing the black-coloured dogs and the Kennel Union of South Africa (KUSA) not accepting these as purebred [4,5] (KUSA, 2008; SABBS, 2019).

Comment

Line 84. "kinship" may be more important than "inbreeding. If a dog has a high inbreeding coefficient but a kinship of O with some other animals, the coefficient can be taken back to O by using that mating. It would be useful, if possible, to include this in the analysis, but this is an optional recommendation.

Answer

Line 84 – Thank you for the recommendation, it will be considered in future studies. Our objective for this study was mainly focused on determining the inbreeding rate and most important ancestors underlying the genetic variability of South African Boerboel dog breed.

Comment

  1. "Bichon Frise" for "Bichon fries"

Answer

Line 303 - These findings are similar to those found for Hannoverian hounds [68] (Voges & Distl, 2009), Bichon frise [24] (Shariflou et al., 2011), Australian shepherd, Papillon, and Phalene [48] (Wijnrocx et al., 2016) dog breeds.